## PERSPECTIVE

# Dynamin independent mechanism of exo-endocytosis coupling

Julia Bandura [ID]
and Zhong-Ping Feng [ID]

*Department of Physiology, Temerty Faculty of Medicine, University of Toronto, Toronto, ON, Canada*

Email: zp.feng@utoronto.ca

Handling Editors: Katalin Toth & Samuel Young

The peer review history is available in the Supporting Information section of this article (https://doi.org/10.1113/JP287403#support-information-section).

Synaptic transmission relies on the synaptic vesicle cycle, in which exocytosis depletes the available pool of vesicles and endocytosis recycles vesicular components and replenishes the readily releasable pool (Wu & Chan, 2024). This recycling process is widely recognized to require the GTPase dynamin, a key player in synaptic vesicle endocytosis. Three mammalian isoforms of dynamin (dynamin 1, 2 and 3) have been identified, and their contributions to endocytosis vary, with some functional redundancy observed (Ferguson & De Camilli, 2012). However, the specific roles of these isoforms in endocytosis and exo-endocytosis coupling remain incompletely understood because of the non-viability of triple dynamin knockout (TKO) animals.

In this issue of *The Journal of Physiology*, Afuwape et al. (2025) show that postnatal depletion of dynamin leads to deficits in multiple but not single vesicle endocytosis. Employing a sophisticated combination of live cell imaging using VGluT1-pHluorin, electron micrography, electrophysiology and immunostaining, Afuwape et al. (2025) have established, for the first time, a postnatal conditional Dnm TKO (Dnm cTKO) genetic mouse model and they probe the effects of total dynamin depletion on evoked multiple vesicle exo-endocytosis in cultured hippocampal neurons following different stimulation intensities, as well as spontaneous single vesicle exo-endocytosis. First, Afuwape et al. (2025) determined that knockout of only one dynamin isoform (2 or 3) did not result in synaptic transmission and synaptic vesicle recycling deficits, consistent with previous work suggesting that dynamins have redundant functions in neurotransmission in mammalian synapses (Ferguson & De Camilli, 2012). Then, Afuwape et al. (2025) used electron microscopy to determine that though triple dynamin knockout leads to a decrease in synaptic vesicle (readily releasable and total) pool size, normal endocytosis persists in more than half of Dnm cTKO synapses following high-frequency stimulation, suggesting that, although dynamins are important for synaptic vesicle replenishment following high-intensity synaptic activity elicited by strong stimulation, there exist dynamin independent mechanisms for maintenance of the synaptic vesicle cycle.

To probe whether dynamins are involved in recycling of single synaptic vesicles, Afuwape et al. (2025) assessed effects of Dnm cTKO on low frequency-induced single vesicle fusion events and found that single synaptic vesicle recycling is unaffected by depletion of dynamins, suggesting that dynamins are only essential for replenishment of the synaptic pool following multiple vesicle release and not single vesicle events. Accordingly, although depletion of dynamin reduced the evoked postsynaptic current amplitude in both excitatory and inhibitory hippocampal neurons, dynamin depletion increased the amplitude and decreased the frequency of miniature postsynaptic currents in both excitatory and inhibitory hippocampal neurons, suggesting at least that dynamin regulates high-frequency evoked and low-frequency evoked/spontaneous neurotransmission via independent mechanisms. Interestingly, dynamin depletion inhibited endocytosis to a greater extent in excitatory glutamatergic synapses than in inhibitory GABAergic synapses, suggesting different dynamin-dependency of endocytosis mechanisms in excitatory synapses compared to inhibitory synapses. This dynamin independent mechanism of single synaptic vesicle recycling was also independent of common components of the clathrin-mediated endocytosis pathway, as assessed by measuring miniature EPSCs in Dnm cTKO hippocampal neurons before and after application of latrunculin, CK-666 and Mdivi-1 to inhibit actin, Arp2/3 and dynamin-related protein 1 (DRP-1), where these inhibitors had no effect on miniature EPSC frequency. Taken together, the results of this study suggest that, although dynamin independent endocytosis occurs to replenish the synaptic vesicle pool sufficiently to sustain single vesicle exocytosis, multiple vesicle availability depends on dynamin-dependent processes (Fig. 1).

The work of Afuwape et al. (2025) provides the first concrete evidence that dynamin independent endocytosis occurs to support single vesicle release, albeit less efficiently than dynamin-dependent processes, whereas dynamin-dependent processes support vesicle availability for multivesicular release. These results provide important insights into potential reinterpretations of previous works focusing on genetic deletions of only one or two isoforms of dynamin, where endocytosis could be compensated by expression of other isoforms, and stimulate discussion of the dynamin dependence of previous conceptualizations of single-vesicle exo-endocytosis processes as identified by application of dynamin inhibitors, such as in kiss-and-run exo-endocytosis in hippocampal neurons (Newton et al., 2006). Furthermore, Afuwape et al. (2025) shows that this mode of dynamin independent single vesicle endocytosis is independent of actin, Arp2/3 and DRP-1 function, suggesting the existence of unidentified molecular mechanisms regulating single vesicle exo-endocytosis coupling beyond the known dynamin-dependent mechanisms.

The mechanisms by which dynamin independent endocytosis contributes to neuronal transmission in mammalian hippocampal neurons have not yet been fully determined, although dynamin independent endocytotic pathways have been reported in other cell types, such as in the calyx of Held (Xu et al., 2008). Exploring how these dynamin independent processes influence neurotransmission remains an intriguing area for future investigation. The current findings also suggest potential differences in synaptic vesicle recycling mechanisms between excitatory and inhibitory synaptic terminals in hippocampal neurons. Future studies may benefit from further examining these differences, particularly in synaptic vesicle

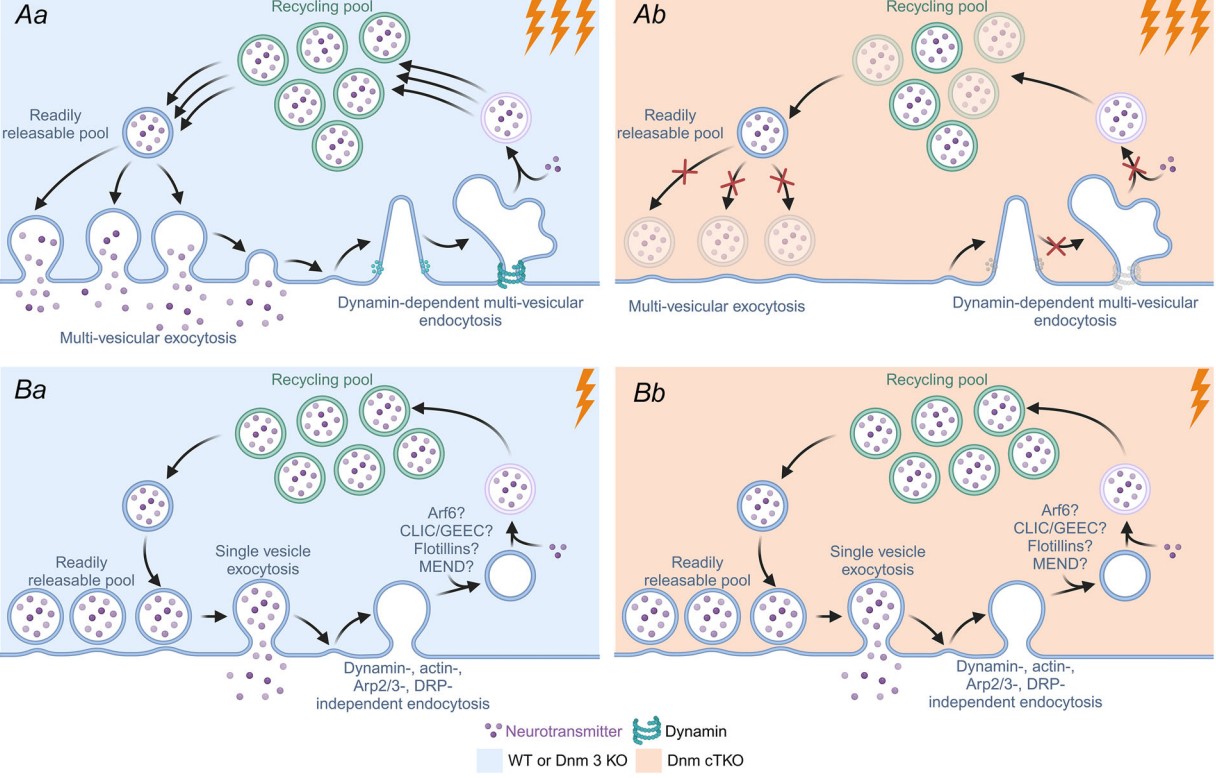

**Figure 1. Dynamin dependency of multivesicular and single vesicle exocytosis**

*Aa*, in wild-type or Dnm 3 KO hippocampal synapses, strong or high-frequency stimulation results in multivesicular exocytosis, supported by dynamin-dependent multivesicle endocytosis that replenishes the recycling vesicle pool and subsequently the readily releasable pool (Wu & Chan, 2024). *Ab*, when dynamins are not present and therefore unavailable for dynamin-dependent multivesicle endocytosis, the readily releasable and recycling pool are depleted, inhibiting multivesicular neurotransmission. *Ba*, wild-type or Dnm 3 KO hippocampal synapses, upon low-intensity stimulation or spontaneously, employ dynamin, actin, Arp2/3 and DRP independent endocytosis mechanisms to replenish the recycling and readily releasable pool of synaptic vesicles sufficiently to support single vesicle release. This process is unaltered by dynamin depletion (*Bb*). Created in BioRender. Bandura, J. (2024) BioRender.com/p79r383.

machinery of excitatory and inhibitory neurons.

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

## Additional information

### Competing interests

No competing interests declared.

### Author contributions

J.B. and Z.P.F. were responsible for the conception or design of the work, as well as drafting the work or revising it critically for important intellectual content. Both authors have approved the final version of the manuscript submitted for publication. Z.P.F. is accountable for all aspects of the work.

### Funding

This work was supported by Canadian Institutes of Health Research (CIHR) (PJT-191824) and Natural Sciences and Engineering Research Council of Canada (NSERC) (RGPIN-2022-04467) to ZPF.

### Keywords

dynamin, endocytosis, Exo-endocytosis

### Supporting information

Additional supporting information can be found online in the Supporting Information section at the end of the HTML view of the article. Supporting information files available:

**Peer Review History**

