## [Peer Review History · The Journal of Physiology]

Dynamin-independent mechanism of exo-endocytosis coupling

Julia Bandura and Zhong-Ping Feng
DOI: 10.1113/JP287403

Corresponding author(s): Zhong-Ping Feng (zp.feng@utoronto.ca)

Review Timeline:

Submission Date:	11-Sep-2024
Accepted:	13-Sep-2024

Senior Editor: Katalin Toth

Reviewing Editor: Samuel Young

Transaction Report:

Dear Dr Feng,

Re: JP-P-2024-287403 "Dynamin-independent mechanism of exo-endocytosis coupling" by Julia Bandura and Zhong-Ping Feng

We are pleased to tell you that your paper has been accepted for publication in The Journal of Physiology.

Authors should note that it is too late at this point to offer corrections prior to proofing. Major corrections at proof stage, such as changes to figures, will be referred to the Editors for approval before they can be incorporated. Only minor changes, such as to style and consistency, should be made at proof stage. Changes that need to be made after proof stage will usually require a formal correction notice.

If you would like to receive our 'Research Roundup', a monthly newsletter highlighting the cutting-edge research published in The Physiological Society's family of journals (The Journal of Physiology, Experimental Physiology and Physiological Reports), please click this link, fill in your name and email address and select 'Research Roundup': <https://www.physoc.org/journals-and-media/membernews/>

Yours sincerely,

Katalin Toth
Senior Editor
The Journal of Physiology

P.S. - You can help your research get the attention it deserves! Check out Wiley's free Promotion Guide for best-practice recommendations for promoting your work at www.wileyauthors.com/eeo/guide. You can learn more about Wiley Editing Services which offers professional video, design, and writing services to create shareable video abstracts, infographics, conference posters, lay summaries, and research news stories for your research at www.wileyauthors.com/eeo/promotion.

IMPORTANT NOTICE ABOUT OPEN ACCESS: To assist authors whose funding agencies mandate public access to published research findings sooner than 12 months after publication, The Journal of Physiology allows authors to pay an Open Access (OA) fee to have their papers made freely available immediately on publication.

You can check if your funder or institution has a Wiley Open Access Account here: <https://authorservices.wiley.com/author-resources/Journal-Authors/licensing-and-open-access/open-access/author-compliance-tool.html>.

EDITOR COMMENTS

Reviewing Editor:

The authors have written an excellent perspective. Thank you.

REFEREE COMMENTS

Referee #1:

This Perspective article accurately presents the article and makes insightful projections about future directions.

Confidential Review

11-Sep-2024